# A differential correction based shadow removal method for real-time monitoring

**Sheng Liu**[1]☯, **Meng Chen**[2]☯, **Zhiheng Li**[3], **Jingxian Liu**[1,4]☯*, **Menglong He**[1]☯

**1** School of Computer Science and Technology, Guangxi University of Science and Technology, Liuzhou, Guangxi, China, **2** Business School, Guilin University of Electronic Technology, Guilin, Guangxi, China, **3** Guangxi LiuGong Machinery Co., Ltd, Liuzhou, Guangxi, China, **4** Liuzhou Key Laboratory of Intelligent Processing and Security of Big Data, Liuzhou, Guangxi, China

☯ These authors contributed equally to this work.
* ljx43031@gxust.edu.cn

## Abstract

Shadow removal is an important issue in the field of motion object surveillance and automatic control. Although many works are concentrated on this issue, the diverse and similar motion patterns between shadows and objects still severely affect the removal performance. Constrained by the computational efficiency in real-time monitoring, the pixel feature based methods are still the main shadow removal methods in practice. Following this idea, this paper proposes a novel and simple shadow removal method based on a differential correction calculation between the pixel values of Red, Green and Blue channels. Specifically, considering the fact that shadows are formed because of the occlusion of light by objects, all the reflected light will be attenuated. Hence there will be a similar weakening trends in all Red, Green and Blue channels of the shadow areas, but not in the object areas. These trends can be caught by differential correction calculation and distinguish the shadow areas from object areas. Based on this feature, our shadow removal method is designed. Experiment results verify that, compared with other state-of-the-art shadow removal methods, our method improves the average of object and shadow detection accuracies by at least 10% in most of the cases.

## Introduction

With the application of intelligent video surveillance and automatic control, shadow removal becomes more and more important. An effective shadow removal method can minimize the interference of shadows on object detection, recognition and control [1, 2]. In fact, shadow, as a phenomenon due to light being blocked by object, has the same motion property with object itself. Therefore, it is difficult to identify and remove shadows based on the judgment of motion property of image areas in the video. Meanwhile, considering the computational efficiency in real-time application and the cost of real-time monitoring equipment, deep neural network based methods [3–7] are difficult to be widely used. Hence, shadow removal is still an interesting and challenge work for real-time monitoring.

**Data Availability Statement:** All the codes and results presented in the study are available from https://github.com/ljx43031/DC-SR-method. Further, this URL can also be found in the last of

The structure of DC-SR method Subsection of this manuscript to support our results.

**Funding:** This work was supported by NSFC under Grant Number 62061003 and Guangxi Science and Technology Plan Project under Grant Numbers AD19245047, in part by Guangxi University of Science and Technology Doctoral Fund under Grant Number XKB19Z02, in part by Liuzhou Science and Technology Plan Project under Grant Numbers 2021ADB0102. All funding was granted to JL. Moreover, this work was also supported in part by Natural Science Foundation of Sichuan Province under Grant Number 2022NSFSC1885. The funder of this project played the role of funding and comparison method provision in this work.

**Competing interests:** The authors have declared that no competing interests exist.

To satisfy the need for real-time computing in object surveillance, background difference based methods [8, 9] are still the most cost-effective methods in practical application. After background difference, all the areas of objects with shadows are detected. Then, the shadows can be further removed based on one or more features which can well distinguish shadows from objects. Currently, the best feature is the RGB pixel ratio [10]. As illustrated in [10], RGB pixel ratios of shadow areas before and after shadow covering are similar, and are significantly different from the ones of object areas. Hence, the shadow areas can be found by RGB pixel ratio comparison (RGB-PRC) method. However, according to the principle of shadow formation, it is the occlusion of light by objects that forms the shadow. In fact, it cannot be sure that the same areas in the images with and without shadows follow the same ratios of RGB pixel values, especially when the light is severely occluded. This phenomenon will be discussed in Section 3. To conquer this problem, this paper investigates all the image data in ISTD dataset [3, 4], and discovers a new pixel feature in the shadow areas of images according to a differential correction calculation. It is named RGB pixel differential correction (RGBP-DC) feature. Further, a new differential correction based shadow removal (DC-SR) is proposed according to the aforementioned RGBP-DC feature. The experiment results show that our DC-SR outperforms the state-of-the-art shadow removal methods.

The organizations of the remainder of this paper are structured as follows. In Related Work Section, the related works are described. In RGB Pixel Differential Correction Feature Section, the limitation of RGB-PRC method is illustrated, and a new RGBP-DC feature is proposed. Then, a new DC-SR method is designed based on the RGBP-DC feature in Differential Correction Based Shadow Removal Method Section. In Experiments Section, a lot of comparison experiments are performed to verify the effectiveness of the proposed DC-SR method. Finally, the conclusion and limitation of the proposed method are given in Conclusion Section. The main contributions of our work are summarized as follows:

- A new pixel feature, i.e., RGBP-DC feature, is found in the shadow areas of images.

- A new differential correction based shadow removal (DC-SR) method is proposed.

## Related work

Currently, in the field of real-time monitoring, we still need the background difference to quickly find out the areas of objects. After that, the goal of shadow removal becomes to distinguish the shadow areas from the object areas based on two kinds of methods: the model-based and feature-based methods, respectively.

Model-based methods mainly use prior information to train corresponding models. For example, Zhang proposed a robust vehicle detection method with shadow elimination [11]. Amin Benish proposed a shadow mask extractor by using a three color attenuation model (TAM) and intensity information to segment the shadow area [12]. Saritha Murali proposed a method to remove shadows from images with uniform textures models [13]. However, those model-based methods depend on the determination of prior information, and also need a lot of training. Hence, the generalization ability of those methods are limited.

Different from model-based methods, feature-based methods mainly concentrate on distinguishing and removing the shadow by contour, brightness, color, texture and other features of pixel which are less affected by environmental factors. Hence, those methods have a wide range of application. For example, Xu obtained the stable shadow elimination results through HSV color features, by using the difference idea of image Log domain [14]. Park used shadow depth map and illumination invariance feature to remove shadows [15].

Li proposed a shadow weakening algorithm based on brightness and texture features without the prior training and manual intervention [16]. Salvador proposed a new cast shadow segmentation algorithm based on the shadow spectrum and geometric characteristics of shadows in the scene [17]. All the performance of shadow removal improved by using one or more features of images, but the computational cost is too expensive to satisfy the real time surveillance for motion object. Tang proposed a low computational cost algorithm to remove shadow according to the differences in foreground and background of the composition of pixel gray feature [18]. Chen further proposed a state-of-the-art shadow removal method: RGB pixel ratio comparison (RGB-PRC) method, based on the similar pixel change features. In this method, the shadow can be distinguished and removed directly according to the ratios of pixels between Red, Green and Blue (RGB) channels in the foreground and background [10]. Therefore, the effect of shadow removal can be greatly improved.

In this paper, we also concentrate on distinguishing and removing the shadow by pixel features. Different from the aforementioned features, the proposed pixel feature is obtained according to both the principle of shadow formation and the statistics of a large number of actual scenes. Hence, the feature proposed in this paper is more typical and has wider applicability. All of the above will be discussed in the next sections.

## RGB pixel differential correction feature

In this section, RGBP-DC feature is discussed, in comparison with the RGB pixel ratio (RGBP-R) feature proposed in [10]. Generally speaking, given a real point $\vec{p}$, let $L(\vec{p})$ and $L_s(\vec{p})$ represent the illuminance reflected from this point with and without the direct light exposures, respectively. In other words, $L_s(\vec{p})$ represents the illuminance reflected from $\vec{p}$ when it is in shadow. Assuming that the coordinate of the corresponding point $\vec{p}$ in imaging plane is $(x, y)$, the pixel values of this point in R, G and B channels are denoted as $R(x, y)$, $G(x, y)$ and $B(x, y)$, respectively, and the ones in shadow are denoted as $R_s(x, y)$, $G_s(x, y)$ and $B_s(x, y)$, respectively. According to [10], the aforementioned RGB pixel ratios $Ratio_R(\cdot)$, $Ratio_G(\cdot)$ and $Ratio_B(\cdot)$ are defined as:

$$Ratio_R(RGB(x, y)) = \frac{R(x, y)}{R(x, y) + G(x, y) + B(x, y)}, \tag{1a}$$

$$Ratio_G(RGB(x, y)) = \frac{G(x, y)}{R(x, y) + G(x, y) + B(x, y)}, \tag{1b}$$

$$Ratio_B(RGB(x, y)) = \frac{B(x, y)}{R(x, y) + G(x, y) + B(x, y)}, \tag{1c}$$

where $RGB(x, y) \triangleq \{R(x, y), G(x, y), B(x, y)\}$.

**Observation 1** *Under the premise that the RGB pixel values are linearly related to the illuminance reflected from $\vec{p}$, the performance of RGB-PRC method can be guaranteed.*

**Analysis 1** *Under this premise in Observation 1, the pixel values of $\vec{p}$ in R, G and B channels can be simply calculated as follows* [17]:

$$R(x, y) = S_R(x, y)L(\vec{p}), \tag{2a}$$

$$G(x, y) = S_G(x, y)L(\vec{p}), \tag{2b}$$

$$B(x, y) = S_B(x, y)L(\vec{p}), \tag{2c}$$

$$R_s(x, y) = S_R(x, y)L_s(\vec{p}), \tag{3a}$$

$$G_s(x, y) = S_G(x, y)L_s(\vec{p}), \tag{3b}$$

$$G_s(x, y) = S_B(x, y)L_s(\vec{p}), \tag{3c}$$

*where $S_R(x, y)$, $S_G(x, y)$ and $S_B(x, y)$ are the linear photoelectric conversion coefficients in R, G and B channels, respectively. Obviously, under this premise of linearity, the ratios of pixel values in R, G and B channels are the same. That is,*

$$Ratio_R(RGB(x, y)) = \frac{S_R(x, y)}{S_R(x, y) + S_G(x, y) + S_B(x, y)} = Ratio_R(RGB_s(x, y)), \tag{4a}$$

$$Ratio_G(RGB(x, y)) = \frac{S_G(x, y)}{S_R(x, y) + S_G(x, y) + S_B(x, y)} = Ratio_G(RGB_s(x, y)), \tag{4b}$$

$$Ratio_B(RGB(x, y)) = \frac{S_B(x, y)}{S_R(x, y) + S_G(x, y) + S_B(x, y)} = Ratio_B(RGB_s(x, y)), \tag{4c}$$

*where $RGB_s(x, y) \triangleq \{R_s(x, y), G_s(x, y), B_s(x, y)\}$. Hence, based on this feature that the RGB pixel ratios with and without the direct light exposure are equal, the shadow area can be distinguish from the object and removed as other background.*

*This completes the analysis of Observation 1.*

However, for most image sensors, the aforementioned linear relationship for imaging is only valid in a certain light intensity range [19, 20]. If light intensity is out of this range, for example the light is severely occluded in strong sunlight, the linear relationship for imaging cannot be guaranteed.

**Observation 2** *In the nonlinear range, the RGBP-R feature no longer exists.*

**Analysis 2** *The nonlinear relationships between RGB pixel values and illuminance reflected from $\vec{p}$ are assumed to be*:

$$R(x, y) = s_r(L(\vec{p}), (x, y)), \tag{5a}$$

$$G(x, y) = s_g(L(\vec{p}), (x, y)), \tag{5b}$$

$$B(x, y) = s_b(L(\vec{p}), (x, y)), \tag{5c}$$

*where $s_r(\cdot)$, $s_g(\cdot)$ and $s_b(\cdot)$ are the nonlinear photoelectric conversion function in R, G and B channels, respectively. To simplify the analysis,* Eq (5) *is linearized based on the Taylor expansion as*

*the following*:

$$R(x, y) = s_r(L_\Delta(\vec{p}), (x, y)) + s'_r(L_\Delta(\vec{p}), (x, y))L_s(\vec{p}) + \circ(L_s(\vec{p})), \tag{6a}$$

$$G(x, y) = s_g(L_\Delta(\vec{p}), (x, y)) + s'_g(L_\Delta(\vec{p}), (x, y))L_s(\vec{p}) + \circ(L_s(\vec{p})), \tag{6b}$$

$$B(x, y) = s_b(L_\Delta(\vec{p}), (x, y)) + s'_b(L_\Delta(\vec{p}), (x, y))L_s(\vec{p}) + \circ(L_s(\vec{p})), \tag{6c}$$

*where $L_\Delta(\vec{p}) = L(\vec{p}) - L_s(\vec{p})$. The reason why $L_\Delta(\vec{p})$ is selected to expand the nonlinear functions is that $L_s(\vec{p})$ is a variable much smaller than $L(\vec{p})$, and the linearized results in (6) can be very closed to the original ones*:

$$R(x, y) \approx s_r(L_\Delta(\vec{p}), (x, y)) + s'_r(L_\Delta(\vec{p}), (x, y))L_s(\vec{p}), \tag{7a}$$

$$G(x, y) \approx s_g(L_\Delta(\vec{p}), (x, y)) + s'_g(L_\Delta(\vec{p}), (x, y))L_s(\vec{p}), \tag{7b}$$

$$B(x, y) \approx s_b(L_\Delta(\vec{p}), (x, y)) + s'_b(L_\Delta(\vec{p}), (x, y))L_s(\vec{p}), \tag{7c}$$

*Similarly, when $\vec{p}$ is in the shadow, the RGB pixel values can be calculated as follows*:

$$R_s(x, y) = s_r(L_s(\vec{p}), (x, y)), \tag{8a}$$

$$G_s(x, y) = s_g(L_s(\vec{p}), (x, y)), \tag{8b}$$

$$B_s(x, y) = s_b(L_s(\vec{p}), (x, y)). \tag{8c}$$

*Further,* Eq (8) *can be linearized based on the Taylor expansion and approximated as the following*:

$$R_s(x, y) \approx s_r(0, (x, y)) + s'_r(0, (x, y))L_s(\vec{p}), \tag{9a}$$

$$G_s(x, y) \approx s_g(0, (x, y)) + s'_g(0, (x, y))L_s(\vec{p}), \tag{9b}$$

$$B_s(x, y) \approx s_b(0, (x, y)) + s'_b(0, (x, y))L_s(\vec{p}). \tag{9c}$$

*For common sensors, the output is 0 when input is 0. Hence,* Eq (9) *can be simplified as follows*:

$$R_s(x, y) \approx s'_r(0, (x, y))L_s(\vec{p}), \tag{10a}$$

$$G_s(x, y) \approx s'_g(0, (x, y))L_s(\vec{p}), \tag{10b}$$

$$B_s(x, y) \approx s'_b(0, (x, y))L_s(\vec{p}). \tag{10c}$$

*Then, according to* (7), *the $Ratio_R(\cdot)$ with direct light from $\vec{p}$ can be approximately calculated as*

$$Ratio_R(RGB(x, y)) \approx \frac{s_r(L_\Delta(\vec{p}), (x, y)) + s'_r(L_\Delta(\vec{p}), (x, y))L_s(\vec{p})}{s_{rgb}(L_\Delta(\vec{p}), (x, y)) + s'_{rgb}(L_\Delta(\vec{p}), (x, y))L_s(\vec{p})}, \tag{11}$$

*where $s_{rgb}(L_\Delta(\vec{p}), (x, y)) = s_r(L_\Delta(\vec{p}), (x, y)) + s_g(L_\Delta(\vec{p}), (x, y)) + s_b(L_\Delta(\vec{p}), (x, y))$,*
*$s'_{rgb}(L_\Delta(\vec{p}), (x, y)) = s'_r(L_\Delta(\vec{p}), (x, y)) + s'_g(L_\Delta(\vec{p}), (x, y)) + s'_b(L_\Delta(\vec{p}), (x, y))$. Moreover,*

*according to (10), the $Ratio_R(\cdot)$ without direct light from $\vec{p}$ can be approximately calculated as*

$$Ratio_R(RGB_s(x,y)) \approx \frac{s'_r(0,(x,y))L_s(\vec{p})}{s'_{rgb}(0,(x,y))L_s(\vec{p})} = \frac{s'_r(0,(x,y))}{s'_{rgb}(0,(x,y))}, \tag{12}$$

*where $s'_{rgb}(0,(x,y)) = s'_r(0,(x,y)) + s'_g(0,(x,y)) + s'_b(0,(x,y))$. Obviously, in most of the cases, $Ratio_R(RGB_s(x,y)) \neq Ratio_R(RGB(x,y))$. In other words, in the R channel, the pixel ratios with and without the direct light exposure are commonly unequal. These issues are also be found in the G and B channels. Hence, in the nonlinear range, the RGBP-R feature no longer exists.*

*This completes the analysis of Observation 2.*

In order to eliminate shadows more effectively and robustly, this paper mines a new image feature, i.e., the RGBP-DC feature, to adapt to most shadow removal situations.

**Observation 3** *The differences between pixel values of point $\vec{p}$ with and without the direct light exposure in R, G and B channels are defined as $\Delta R(x,y)$, $\Delta G(x,y)$ and $\Delta B(x,y)$, respectively. Under a stable monitoring scenario, there are stable linear relationships on $\Delta R(x,y)$, $\Delta G(x,y)$ and $\Delta B(x,y)$.*

**Analysis 3** *Under the stable monitoring scenario, the reduced illuminance $L_\Delta(\vec{p})$ caused by the occlusion of light by different objects in different space are similar. Hence, $L_\Delta(\vec{p})$ can be approximately replaced by a constant value C.*

*In Eq (7), Hence, Eq (7) can be further simplified as follows:*

$$R(x,y) \approx s_r(C,(x,y)) + s'_r(C,(x,y))L_s(\vec{p}), \tag{13a}$$

$$G(x,y) \approx s_g(C,(x,y)) + s'_g(C,(x,y))L_s(\vec{p}), \tag{13b}$$

$$B(x,y) \approx s_b(C,(x,y)) + s'_b(C,(x,y))L_s(\vec{p}). \tag{13c}$$

*Then, jointing Eq (10), the differences of pixel values of points with and without the direct light exposure in different channels are calculated as follows:*

$$\Delta R(x,y) = R(x,y) - R_s(x,y) \approx s_r(C,(x,y)) + (s'_r(C,(x,y) - s'_r(0,(x,y)))L_s(\vec{p}), \tag{14a}$$

$$\Delta G(x,y) = G(x,y) - G_s(x,y) \approx s_g(C,(x,y)) + (s'_g(C,(x,y) - s'_g(0,(x,y)))L_s(\vec{p}), \tag{14b}$$

$$\Delta B(x,y) = B(x,y) - B_s(x,y) \approx s_b(C,(x,y)) + (s'_b(C,(x,y) - s'_b(0,(x,y)))L_s(\vec{p}). \tag{14c}$$

*Because $s_r(C,(x,y))$, $s_g(C,(x,y))$, $s_b(C,(x,y))$, $s'_r(C,(x,y))$, $s'_g(C,(x,y))$, $s'_b(C,(x,y))$, $s'_r(0,(x,y))$, $s'_g(0,(x,y))$ and $s'_b(0,(x,y))$ are all unknown constants in the stable monitoring scenarios. The stable relationships between $\Delta R(x,y)$, $\Delta G(x,y)$ and $\Delta B(x,y)$ can be obtained by*

*eliminating $L_s(\vec{p})$ as follows:*

$$\Delta R(x,y) \approx s_r(C,(x,y)) + \frac{(s'_r(C,(x,y)) - s'_r(0,(x,y)))}{(s'_b(C,(x,y)) - s'_b(0,(x,y)))}(\Delta B(x,y) - s_b(C,(x,y))), \qquad (15a)$$

$$\Delta G(x,y) \approx s_g(C,(x,y)) + \frac{(s'_g(C,(x,y)) - s'_g(0,(x,y)))}{(s'_b(C,(x,y)) - s'_b(0,(x,y)))}(\Delta B(x,y) - s_b(C,(x,y))), \qquad (15b)$$

$$\Delta B(x,y) \approx s_b(C,(x,y)) + \frac{(s'_b(C,(x,y)) - s'_b(0,(x,y)))}{(s'_g(C,(x,y)) - s'_g(0,(x,y)))}(\Delta G(x,y) - s_g(C,(x,y))). \qquad (15c)$$

Eq (15) *can be further simplified as follows:*

$$\Delta R(x,y) \approx M_{B2R} + N_{B2R}\Delta B(x,y), \qquad (16a)$$

$$\Delta G(x,y) \approx M_{B2G} + N_{B2G}\Delta B(x,y), \qquad (16b)$$

$$\Delta R(x,y) \approx M_{G2R} + N_{G2R}\Delta G(x,y), \qquad (16c)$$

*where*

$$N_{B2R} = \frac{(s'_r(C,(x,y)) - s'_r(0,(x,y)))}{(s'_b(C,(x,y)) - s'_b(0,(x,y)))}, \qquad (17a)$$

$$M_{B2R} = s_r(C,(x,y)) - N_{B2R}s_b(C,(x,y)), \qquad (17b)$$

$$N_{B2G} = \frac{(s'_g(C,(x,y)) - s'_g(0,(x,y)))}{(s'_b(C,(x,y)) - s'_b(0,(x,y)))}, \qquad (17c)$$

$$M_{B2G} = s_g(C,(x,y)) - N_{B2G}s_b(C,(x,y)), \qquad (17d)$$

$$N_{G2R} = \frac{(s'_r(C,(x,y)) - s'_r(0,(x,y)))}{(s'_g(C,(x,y)) - s'_g(0,(x,y)))}, \qquad (17e)$$

$$M_{G2R} = s_r(C,(x,y)) - N_{G2R}s_g(C,(x,y)). \qquad (17f)$$

*Obviously, $N_{B2R}$, $M_{B2R}$, $N_{B2G}$, $M_{B2G}$, $N_{G2R}$ and $M_{G2R}$ are all unknown constants. Hence, under a stable monitoring scenario, all these aforementioned constants can be calculated by known $\Delta R(x, y)$, $\Delta G(x, y)$ and $\Delta B(x, y)$ in advance. Then, the stable linear relationships between the pixel differential values in R, G and B channels can be derived.*

*This completes the analysis of Observation 3.*

Therefore, under the linear correction with Eq (16), the differences of $\Delta R(x, y)$, $\Delta G(x, y)$ and $\Delta B(x, y)$ are very small. That is:

$$\Delta\Delta R\&B(x,y) = \Delta R(x,y) - [M_{B2R} + N_{B2R}\Delta B(x,y)] \rightarrow 0, \qquad (18a)$$

$$\Delta\Delta G\&B(x,y) = \Delta G(x,y) - [M_{B2G} + N_{B2G}\Delta B(x,y)] \rightarrow 0, \qquad (18b)$$

$$\Delta\Delta R\&G(x,y) = \Delta R(x,y) - [M_{G2R} + N_{G2R}\Delta G(x,y)] \rightarrow 0. \qquad (18c)$$

Given any small threshold $T$, it is easily to find out that:

$$\Delta\Delta R\&B(x, y) < T, \tag{19a}$$

$$\Delta\Delta G\&B(x, y) < T, \tag{19b}$$

$$\Delta\Delta R\&G(x, y) < T. \tag{19c}$$

This is the RGBP-DC feature, which can be used to discover and remove the shadow areas.

## Differential correction based shadow removal method

In this section, the proposed DC-SR method is described in detail. Firstly, based on ISTD dataset [3, 4], a set of parameters in Eq (18) is determined for surveillance environments under common daylight. Secondly, the structure of shadow removal method is designed and the algorithmic complexity is discussed.

### Parameters estimation according to ISTD dataset

As seen in Eq (16), constants $N_{B2R}$, $M_{B2R}$, $N_{B2G}$, $M_{B2G}$, $N_{G2R}$ and $M_{G2R}$ can be learnt as the unknown parameters, given known $\Delta R(x, y)$, $\Delta G(x, y)$ and $\Delta B(x, y)$ under actual monitoring scenes.

A major surveillance scene is monitoring during the day or under sunlight lamps. The light source for this monitoring is sunlight. This paper uses the ISTD dataset, in which all of images are taken under sunlight, to estimate those unknown parameters in such scene. Specifically, as seen in Fig 1, there are three kinds of image in each triplet of ISTD dataset: shadow image, shadow mask image and shadow-free image. To obtain a stable relationships between $\Delta R(x, y)$, $\Delta G(x, y)$ and $\Delta B(x, y)$, this paper derives the values of $N_{B2R}$, $M_{B2R}$, $N_{B2G}$, $M_{B2G}$, $N_{G2R}$ and $M_{G2R}$ based on the statistics of all triplets in this database.

First, for the $i$th triplet, the $\Delta\bar{R}(i)$, $\Delta\bar{G}(i)$ and $\Delta\bar{B}(i)$, i.e., the means of all differences between the pixel values without and with shadow in shadow area of each R, G and B channel are calculated as:

$$\Delta\bar{R}(i) = \frac{1}{N}\sum_{p \in P_s} R_{sf}(i, p)\min[M(i, p), 1] - R_s(i, p)\min[M(i, p), 1], \tag{20a}$$

$$\Delta\bar{G}(i) = \frac{1}{N}\sum_{p \in P_s} G_{sf}(i, p)\min[M(i, p), 1] - G_s(i, p)\min[M(i, p), 1], \tag{20b}$$

$$\Delta\bar{B}(i) = \frac{1}{N}\sum_{p \in P_s} B_{sf}(i, p)\min[M(i, p), 1] - B_s(i, p)\min[M(i, p), 1], \tag{20c}$$

where $P_s$ is the pixel set of images of the $i$th triplet, $p$ is the pixel in $P_s$, $N$ is the number of pixels in $P_s$. $R_{sf}(i, p)$, $G_{sf}(i, p)$, $B_{sf}(i, p)$, $R_s(i, p)$, $G_s(i, p)$ and $B_s(i, p)$ are the values of $p$ in R, G and B channels of shadow-free image and shadow image, respectively. $M(i, p)$ is the value of $p$ in shadow mask image.

To simplify the expression of means of differences, this paper uses $\Delta\bar{R}$, $\Delta\bar{G}$ and $\Delta\bar{B}$ as common notations for the means of differences of any triplet. As seen in Eq (16), there are linear relationships between $\Delta\bar{R}$, $\Delta\bar{G}$ and $\Delta\bar{B}$. Obviously, $|\Delta\bar{R}|$, $|\Delta\bar{G}|$ and $|\Delta\bar{B}|$ will also obey linear relationships, further obey the RGBP-DC feature. To simplify subsequent calculations, this

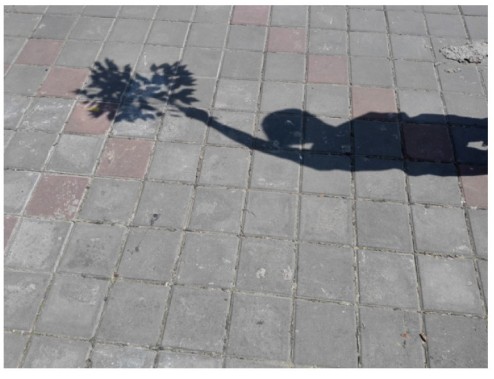
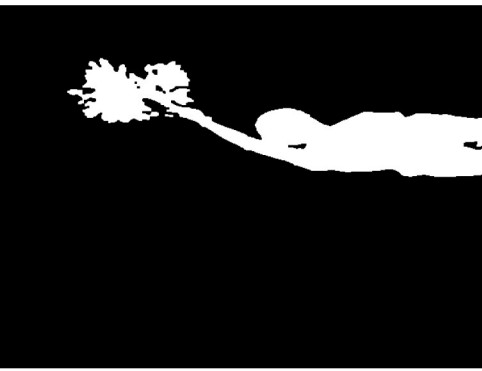

(a) shadow image (b) shadow mask image

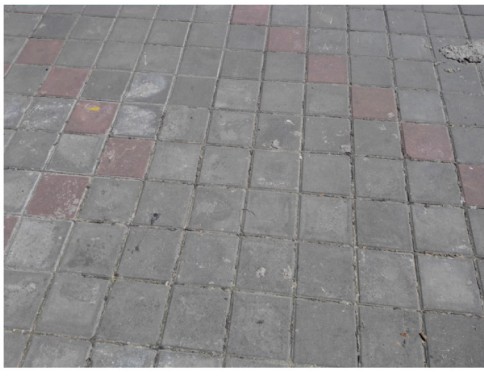

(c) shadow-free image

**Fig 1. Images of one triplet in ISTD dataset.** (a) shadow image. (b) shadow mask image. (c) shadow-free image.

paper analyses the relationships of $|\Delta\bar{R}|$, $|\Delta\bar{G}|$ and $|\Delta\bar{B}|$ to obtain the RGBP-DC feature, instead of the ones of $\Delta\bar{R}$, $\Delta\bar{G}$ and $\Delta\bar{B}$.

The relationship of $|\Delta\bar{R}|$, $|\Delta\bar{G}|$ and $|\Delta\bar{B}|$ of all the triplet in ISTD dataset is summarized in Fig 2. In this figure, each red points in three sub-figures are the pairs of $(|\Delta\bar{B}|, |\Delta\bar{G}|)$, $(|\Delta\bar{B}|, |\Delta\bar{R}|)$ and $(|\Delta\bar{G}|, |\Delta\bar{R}|)$ of the $i$ triplet. Obviously, the linear relationships can be fitted as the blue lines in Fig 2, and their function is shown as follows,

$$| \Delta\bar{R} | = 0.39 + 1.26 | \Delta\bar{B} |, \tag{21a}$$

$$| \Delta\bar{G} | = 0.85 + 1.10 | \Delta\bar{B} |, \tag{21b}$$

$$| \Delta\bar{R} | = 0.11 + 1.11 | \Delta\bar{G} | . \tag{21c}$$

Hence, according Eq (21), the RGBP-DC feature for common daylight monitoring is derived.

## The structure of DC-SR method

Generally speaking, the goal of shadow removal for monitoring is to eliminate the effect of shadows on object recognition. When objects are detected by cameras, the shadows are also detected as part of the objects, thus seriously affecting the accuracy of detection. Because there is no RGBP-DC feature in the actual object areas of foreground, the shadows areas can be found and distinguished from object areas by RGBP-DC feature.

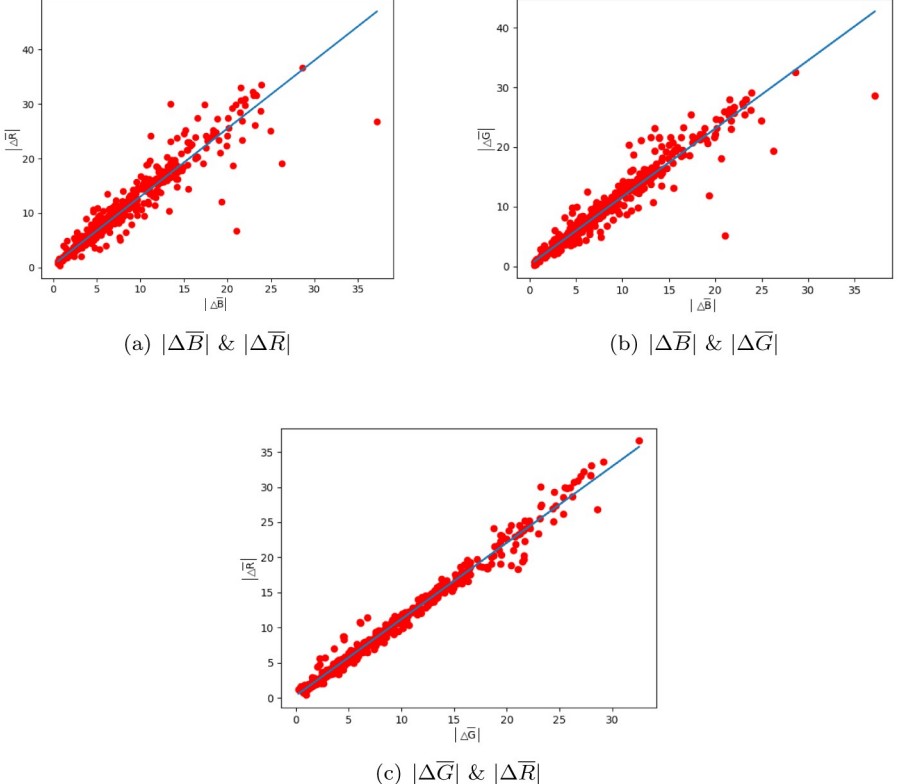

(a) $|\Delta\overline{B}|$ & $|\Delta\overline{R}|$

(b) $|\Delta\overline{B}|$ & $|\Delta\overline{G}|$

(c) $|\Delta\overline{G}|$ & $|\Delta\overline{R}|$

**Fig 2. The relationships of $|\Delta\overline{R}|$, $|\Delta\overline{G}|$ and $|\Delta\overline{B}|$.** (a) $|\Delta\overline{B}|$ & $|\Delta\overline{R}|$. (b) $|\Delta\overline{B}|$ & $|\Delta\overline{G}|$. (c) $|\Delta\overline{G}|$ & $|\Delta\overline{R}|$.

Specifically, based on the RGBP-DC feature, the structure of the proposed DC-SR method is described in Fig 3. As we can see in this figure, a background image $I_b$ need to be firstly determined before monitoring. Then, given a foreground image $I_f$, the absolute value of the first difference of the complete image $\Delta I_c$ is calculated as follows,

$$\mid \Delta I_c \mid = \mid I_b - I_f \mid . \tag{22}$$

Obviously, $|\Delta I_c|$ includes R, G and B channels, i.e., $|\Delta R_c|$, $|\Delta G_c|$ and $|\Delta B_c|$. In addition, the grayscales of foreground and background are calculated, and the mask of objects with shadow in foreground image, which is denoted as $I_m$, is derived by thresholding the absolute differences of grayscales. Then, the first order differences of objects with shadow in R, G and B channels are calculated as follows,

$$\mid \Delta R \mid = \mid \Delta R_c \mid I_m, \tag{23a}$$

$$\mid \Delta G \mid = \mid \Delta G_c \mid I_m, \tag{23b}$$

$$\mid \Delta B \mid = \mid \Delta B_c \mid I_m. \tag{23c}$$

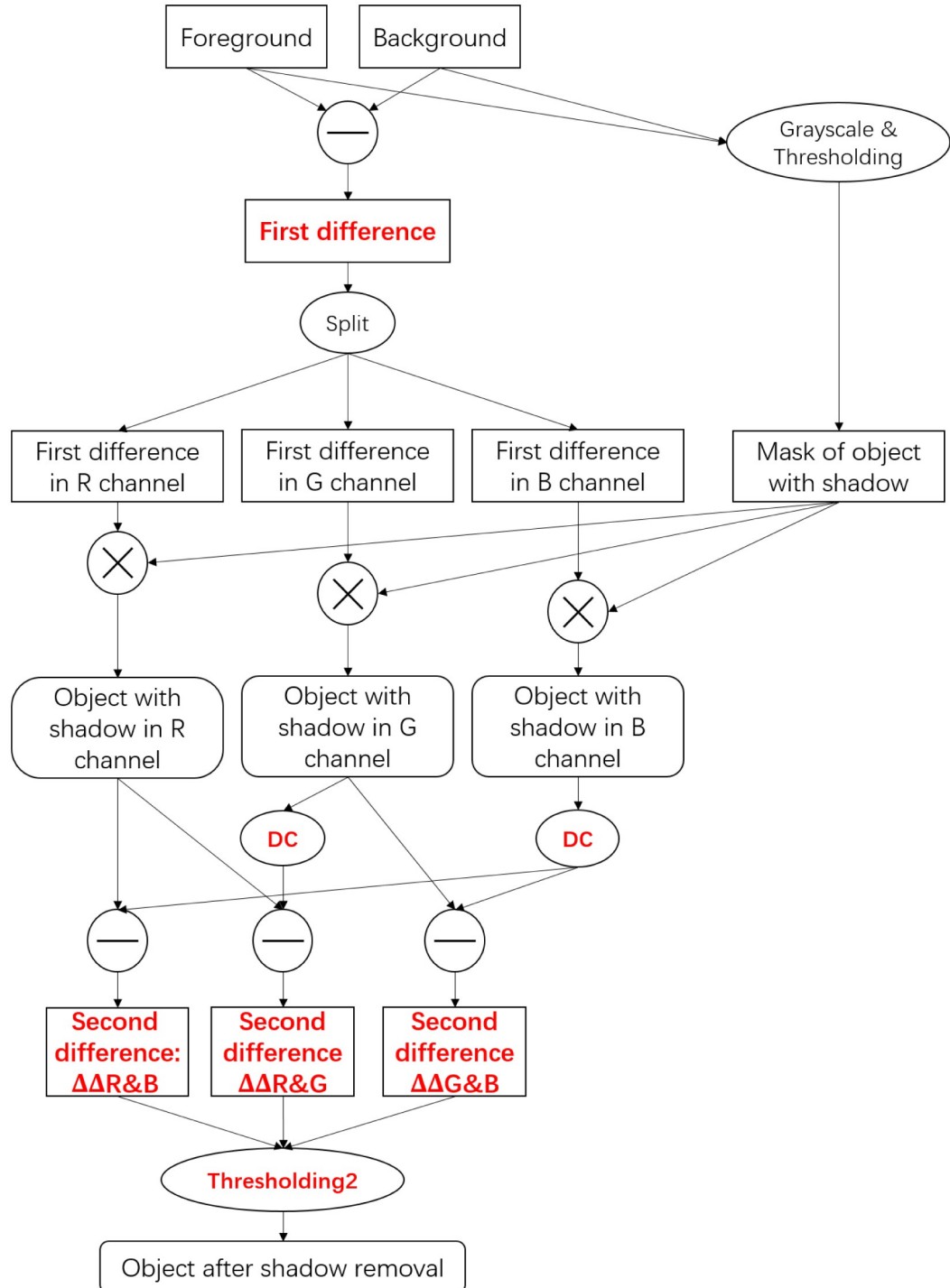

**Fig 3. The calculation structure of DC-SR.**

According to the RGBP-DC feature, the second differences are calculated by differential correction (DC) Eq (21) as follows,

$$\Delta\Delta\boldsymbol{R}\&\boldsymbol{B} =\mid \Delta\boldsymbol{R} \mid -[0.39 + 1.26 \mid \Delta\boldsymbol{B} \mid],\tag{24a}$$

$$\Delta\Delta\boldsymbol{G}\&\boldsymbol{B} =\mid \Delta\boldsymbol{G} \mid -[0.85 + 1.10 \mid \Delta\boldsymbol{B} \mid],\tag{24b}$$

$$\Delta\Delta\boldsymbol{R}\&\boldsymbol{G} =\mid \Delta\boldsymbol{R} \mid -[0.11 + 1.11 \mid \Delta\boldsymbol{G} \mid].\tag{24c}$$

A proper constant $T$ is set in "Thresholding2" to distinguish the shadow as follows:

$$\Delta\Delta\boldsymbol{R}\&\boldsymbol{B}(x, y) < T,\tag{25a}$$

$$\Delta\Delta\boldsymbol{G}\&\boldsymbol{B}(x, y) < T,\tag{25b}$$

$$\Delta\Delta\boldsymbol{R}\&\boldsymbol{G}(x, y) < T.\tag{25c}$$

Then any pixel in image, which satisfies (25), is considered to be shadow pixel and removed. That is, in "Thresholding2", all pixels satisfy inequalities (25) are set to be 0, and others are set to be 255. Then the image of object after shadow removal can be derived. The whole calculation process of this method is summarized in Algorithm 1. All the codes and results can be found at: https://github.com/ljx43031/DC-SR-method.

**Algorithm 1** DC-SR method

```
Input: Foreground image I_f, background image I_b, the values of N_{B2R},
M_{B2R}, N_{B2G}, M_{B2G}, N_{G2R} and M_{G2R}, threshold T.
Output: The binary image of objects without shadows.
  1. |ΔI_c| is calculated as: |ΔI_c| = |I_b - I_f|.
  2. Grayscale I_f and I_b to get I_{fg} and I_{bg}.
  3. Thresholding |I_{fg} - I_{bg}| → I_m
  4. |ΔI_c[:, :, 2]| I_m → |ΔR|
  5. |ΔI_c[:, :, 1]| I_m → |ΔG|
  6. |ΔI_c[:, :, 0]| I_m → |ΔB|
  7. ΔR - [M_{B2R} + N_{B2R}ΔB] → ΔΔR&B
  8. ΔG - [M_{B2G} + N_{B2G}ΔB] → ΔΔG&B
  9. ΔR - [M_{G2R} + N_{G2R}ΔG] → ΔΔR&G
 10. For: pixel p(x, y) in image:
   (a) If:ΔΔR&B(x, y) < T and ΔΔG&B(x, y) < T and ΔΔR&G(x, y) < T
   The value of p(x, y) is set to be 255
   (b) Else:
   The value of p(x, y) is set to be 0
   End
```

## Time complexity analysis

As described in Algorithm 1, there are 10 steps for each shadow removal calculation. Assuming the image size is $N \times M \times 3$, step 1 contains $N \times M \times 3$ subtractions and absolute value calculations, hence the time complexity is $\boldsymbol{O}(N \times M \times 6)$. Step 2 needs to grayscale $\boldsymbol{I}_f$ and $\boldsymbol{I}_b$, which in fact averages the pixel values $\boldsymbol{I}_f$ and $\boldsymbol{I}_b$. Hence, this calculation performs two additions and one division for each pixel, and the time complexity is $\boldsymbol{O}(N \times M \times 6)$. Step 3 contains $N \times M$ subtractions and thresholding calculations, hence the time complexity is $\boldsymbol{O}(N \times M \times 2)$. Moreover, it can be easily known that the time complexity is $\boldsymbol{O}(N \times M \times 3)$ from step 4 to 6, and $\boldsymbol{O}(N \times M \times 9)$ from step 7 to 9. Step 10 is the judgements for each pixel, whose time complexity is $\boldsymbol{O}(N \times M \times 3)$. Obviously, the total time complexity of this algorithm is $\boldsymbol{O}(N \times M \times 29)$. In other

words, the time complexity of this method is linearly related to the number of pixels of the video frame.

## Experiments

In this section, our DC-SR method is evaluated in both outdoor and indoor (with sunlight lamp) environments, in comparison with RGB-PRC method and the Gray Levels Comparison (GLC) method [18]. Further, we test our DC-SR method under real-time monitoring, in order to prove its reliability and stability. For a fair comparison, the thresholds used in all the aforementioned methods are fixed. Specifically, the threshold used in our DC-SR method is set to be 8. According to [10, 18], the thresholds used in RGB-PRC and GLC method are set to be 0.008 and 35, respectively.

### The implementation description

In this paper, we use Hikvision 2 megapixel USB camera to take photos and use ordinary computer to run the program of the proposed method.

### Analysis of shadow removal performance in static scene

**Outdoor environment.**  We compare the shadow removal performances of the aforementioned three methods in outdoor environment. The results are shown in Fig 4. In this figure, the first column shows the background image, the second column shows the foreground image, the third column shows the foreground image with object circled in red lines, the forth to sixth columns show the shadow removal results of the GLC, RGB-PRC and DC-SR methods, respectively. In this bright outdoor environment, the backlit sides of the objects are very dark, which are very similar as the shadows in terms of the intensity of light reflection. Hence, those methods, such as the GLC method, which distinguish shadows relying on the intensity of light reflections will fail. This problem can be obviously seen in the forth column of Fig 4. Moreover, as mentioned in RGB Pixel Differential Correction Feature Section, the RGBP-R feature is not accurate enough to distinguish the shadow area from the object area in those bright light environments. Hence, as seen in the fifth column of Fig 4, the shadow removal performance of RGB-PRC method degrades in those environments. That is, if the object can be completely detected, the shadow cannot be perfectly removed, for example the image in the third row and fifth column. Conversely, if the shadow can be perfectly removed, the object cannot be completely detected, for example the image in the sixth row and fifth column. However, the shadow removal results of the sixth column of Fig 4 show that the proposed DC-SR method can accurately detected the object while well removing the shadow. Hence, our DC-SR method can outperforms other shadow removal methods in outdoor environment with bright light.

**Indoor environment.**  As seen in Fig 5, both the performance of GLC and RGB-PRC are improved because the light intensity is much weaker than sunlight. But obviously, the proposed DC-SR method still provides the most accurate object detection results with similar shadow removal performances.

**Evaluation metric.**  To further verify the shadow removal effect of DC-SR method, we propose the average of object and shadow detection accuracies as the evaluation metric. specifically, the essence of shadow removal is to distinguish the shadows from objects. In other words, the objects need to be correctly detected while well removing the shadows. Hence, we average the object and shadow detection accuracies to obtain a proper overall merit for shadow

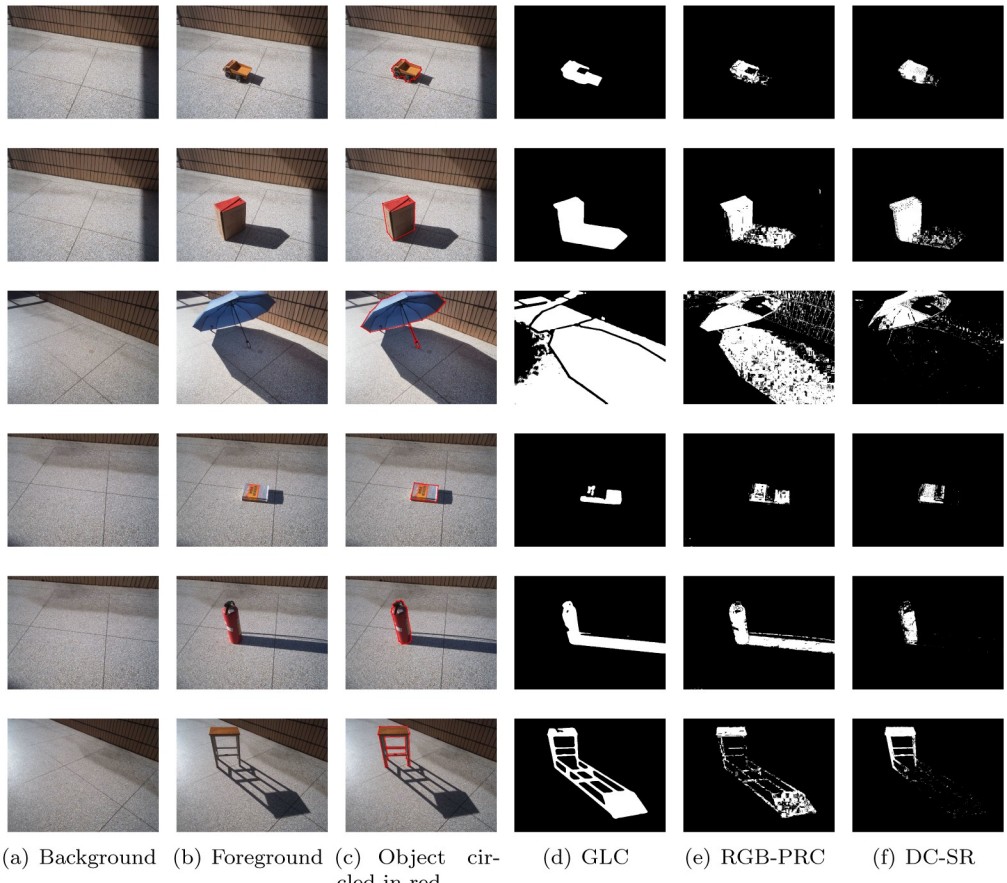

(a) Background  (b) Foreground  (c) Object circled in red  (d) GLC  (e) RGB-PRC  (f) DC-SR

**Fig 4. Comparison of shadow removal with GLC, RGB-PRC and DC-SR methods, respectively, in outdoor environments.** (a) Background. (b) Foreground. (c) Object circled in red. (d) GLC. (e) RGB-PRC. (f) DC-SR.

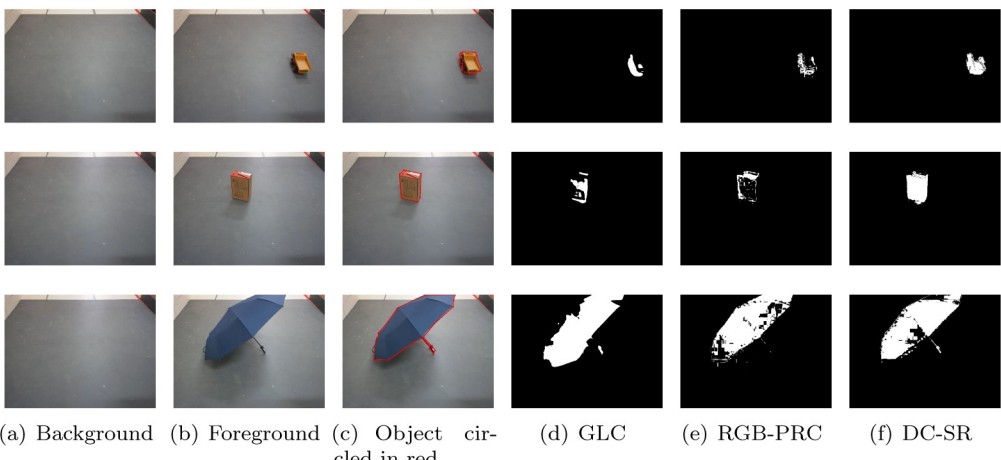

(a) Background  (b) Foreground  (c) Object circled in red  (d) GLC  (e) RGB-PRC  (f) DC-SR

**Fig 5. Comparison of shadow removal with GLC, RGB-PRC and DC-SR methods, respectively, in indoor environments.** (a) Background (b) Foreground (c) Object circled in red. (d) GLC. (e) RGB-PRC. (f) DC-SR.

removal as follows:

$$\frac{1}{2}\left(\frac{N_{object}^{detected}}{N_{object}^{actual}} + \frac{N_{shadow}^{detected}}{N_{shadow}^{actual}}\right), \tag{26}$$

where $N_{object}^{detected}$ and $N_{shadow}^{detected}$ are the numbers of pixels of detected object and shadow areas, respectively. $N_{object}^{actual}$ and $N_{shadow}^{actual}$ are the numbers of pixels of actual object and shadow areas, respectively. To use the metric (26), actual object and shadow areas need to be known first. Hence, we manually marked the counters of the object for each case, as seen in the third columns of Figs 4 and 5, to get the actual object area. Further, we eliminate the actual object area from the difference image of foreground and background to obtain the actual shadow area. The comparison results are shown in Table 1. The cases of outdoors 1 to 6 corresponds to each row of Fig 4 and the cases of indoors 1 to 3 corresponds to each row of Fig 5. In Table 1, we can see that, in each case, the proposed DC-SR method improves the average of object and shadow detection accuracies by at least 10% except Indoors 3. But in fact, the average accuracy of proposed method is still higher that other methods in the Indoors 3 case. Hence, our DC-SR method outperforms other state-of-the-art shadow removal methods.

**Table 1. Accuracy comparison.**

| Cases | | GLC | RGB-PRC (m) | DC-SR |
|---|---|---|---|---|
| **Outdoors 1** | Object detection accuracy | 0.80 | 0.65 | 0.86 |
| | Shadow removal accuracy | 0.45 | 0.82 | 0.80 |
| | Average accuracy | 0.63 | 0.73 | 0.83 |
| **Outdoors 2** | Object detection accuracy | 0.89 | 0.88 | 0.93 |
| | Shadow removal accuracy | 0.50 | 0.70 | 0.87 |
| | Average accuracy | 0.69 | 0.79 | 0.90 |
| **Outdoors 3** | Object detection accuracy | 0.64 | 0.85 | 0.74 |
| | Shadow removal accuracy | 0.32 | 0.65 | 0.99 |
| | Average accuracy | 0.48 | 0.75 | 0.86 |
| **Outdoors 4** | Object detection accuracy | 0.38 | 0.56 | 0.64 |
| | Shadow removal accuracy | 0.52 | 0.63 | 0.92 |
| | Average accuracy | 0.45 | 0.59 | 0.78 |
| **Outdoors 5** | Object detection accuracy | 0.85 | 0.83 | 0.61 |
| | Shadow removal accuracy | 0.47 | 0.48 | 0.99 |
| | Average accuracy | 0.66 | 0.66 | 0.80 |
| **Outdoors 6** | Object detection accuracy | 0.75 | 0.51 | 0.78 |
| | Shadow removal accuracy | 0.43 | 0.73 | 0.97 |
| | Average accuracy | 0.59 | 0.62 | 0.87 |
| **Indoors 1** | Object detection accuracy | 0.37 | 0.44 | 0.83 |
| | Shadow removal accuracy | 0.76 | 0.92 | 0.89 |
| | Average accuracy | 0.56 | 0.68 | 0.86 |
| **Indoors 2** | Object detection accuracy | 0.35 | 0.30 | 0.92 |
| | Shadow removal accuracy | 0.95 | 0.95 | 0.91 |
| | Average accuracy | 0.65 | 0.63 | 0.92 |
| **Indoors 3** | Object detection accuracy | 0.79 | 0.77 | 0.81 |
| | Shadow removal accuracy | 0.83 | 0.98 | 0.98 |
| | Average accuracy | 0.81 | 0.88 | 0.90 |

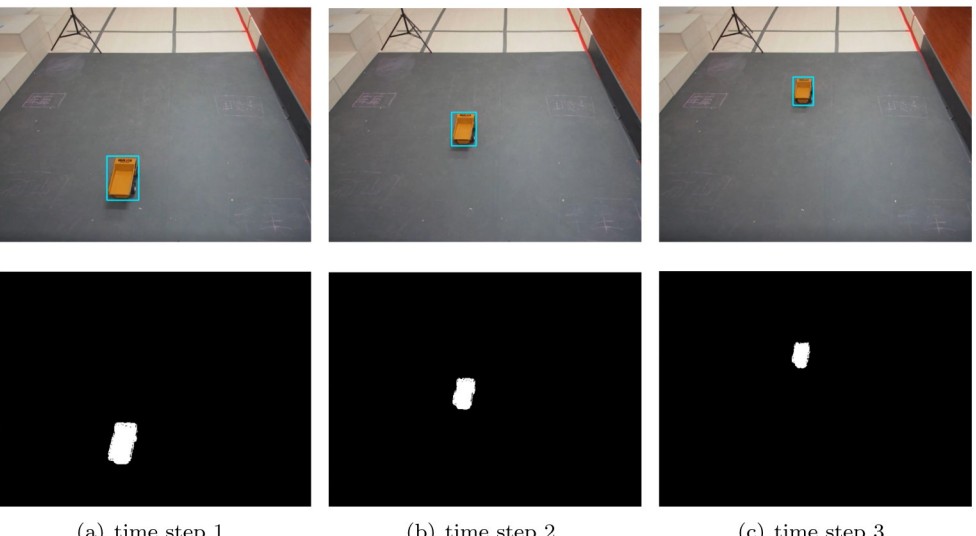

**Fig 6. Shadow removal results at different time steps with our DC-SR method under real-time monitoring, when object moves in a straight line.** (a) time step 1. (b) time step 2. (c) time step 3.

## Performance testing for monitoring

To further verify the performance of our DC-SR method for monitoring, we test our DC-SR method under real-time monitoring. The results are shown in Figs 6 and 7. In the first rows of the two figures, the blue tracking boxes correctly frame the object area without the shadow area at any time steps. Moreover, in the second rows of the two figures, the white areas are the objects detected after shadow removal. Obviously, we can see that no shadow areas are included and the car was correctly detected.

Meanwhile, we test the computational time of all the aforementioned methods with the same Intel Core i7-8700 CPU at 3.2 GHz and 32 GB RAM. The results are that the proposed

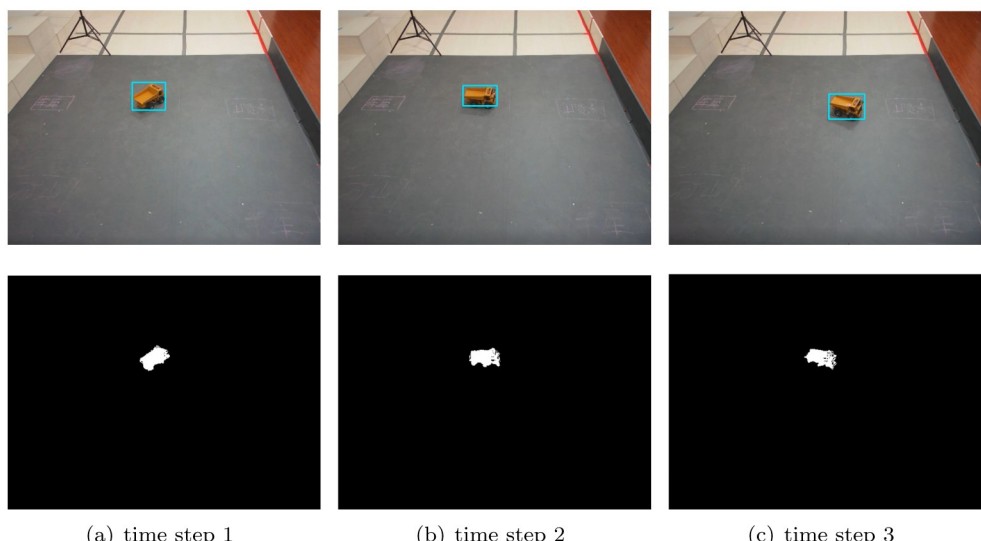

**Fig 7. Shadow removal results at different time steps with our DC-SR method under real-time monitoring, when object turns.** (a) time step 1. (b) time step 2. (c) time step 3.

DC-SR method consumes 19.9ms for each removal calculation while the GLC and RGB-PRC methods consume 5.0 and 20.9 ms, respectively. Hence, the computational efficiency of the proposed DC-SR method meets the requirements of monitoring, which can be further verified that no frame drops were found in real-time monitoring experiments. Therefore, our DC-SR method can achieve efficient and accurate shadow removal in real-time monitoring.

## Conclusion

In this paper, we propose a new differential correction based shadow removal (DC-SR) method based on the new RGB pixel differential correction (RGBP-DC) feature in the shadow areas. From the effect of shadow removal, the proposed RGBP-DC feature, which can well distinguish the shadow areas from objects, is more suitable for shadow removal under both daylight and sunlight lamp environments. Experiments proves that our DC-SR method performs better in comparison with the state-of-the-art shadow removal methods of monitoring. Further, the results of time complexity analysis and algorithm testing in real-time monitoring show that our DC-SR method has the ability to efficiently and accurately remove shadows.

In fact, the performance of our DC-SR method is closely related to the parameters in (16). Although those parameters are set based on the ISTD dataset which covers the main daylight environments and represents the most common relationship between shadow and shadow-free images, in some special low-light or polarized environments, the performance of our method will still degrade. How to improve the adaptability of the method to those special environments will be an important future work.

## Author Contributions

**Data curation:** Menglong He.

**Resources:** Zhiheng Li.

**Writing – original draft:** Sheng Liu, Meng Chen, Jingxian Liu.

**Writing – review & editing:** Meng Chen, Jingxian Liu.

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
