## [Decision Letter · Decision Letter 0]

2 Aug 2022

PONE-D-22-17507A differential correction based shadow removal method for real-time monitoringPLOS ONE

Dear Dr. Liu,

Thank you for submitting your manuscript to PLOS ONE. After careful consideration, we feel that it has merit but does not fully meet PLOS ONE’s publication criteria as it currently stands. Therefore, we invite you to submit a revised version of the manuscript that addresses the points raised during the review process.

We look forward to receiving your revised manuscript.

Kind regards,

Zhaoqing Pan, Ph.D.

Academic Editor

PLOS ONE

Journal Requirements:

"This work was supported by NSFC under Grant Number 62061003 and Guangxi Science and Technology Project under Grant Numbers AD19245047, in part by Guangxi University of Science and Technology Doctoral Fund under Grant Number XKB19Z02, in part by Liuzhou Science and Technology Plan Project under Grant Numbers 2021ADB0102, in part by Natural Science Foundation of Guangxi Provence under 

Grant Numbers 2019GXNSFAA245049".

"This work was supported by NSFC under Grant Number 62061003 and Natural Science Foundation of Guangxi Provence under Grant Numbers AD19245047 and 2019GXNSFAA245049, in part by Guangxi University of Science and Technology Doctoral Fund under Grant Number XKB19Z02, in part by Liuzhou Science and Technology Plan Project under Grant Numbers 2021ADB0102".

Additional Editor Comments:

This submission needs a major revision.

Reviewers' comments:

Reviewer's Responses to Questions

**Comments to the Author**

1. Is the manuscript technically sound, and do the data support the conclusions?

Reviewer #1: Partly

2. Has the statistical analysis been performed appropriately and rigorously? 

Reviewer #1: No

3. Have the authors made all data underlying the findings in their manuscript fully available?

Reviewer #1: No

4. Is the manuscript presented in an intelligible fashion and written in standard English?

Reviewer #1: Yes

5. Review Comments to the Author

Reviewer #1: ID : PONE-D-22-17507

Title : A differential correction based shadow removal method for real-time monitoring

Summary:

In this work, a shadow removal method-based on a differential correction is proposed. The proposed method computes the differential correlation between the pixel values of the channel in RGB color space.

The manuscript is interesting; however, the following comment should be addressed :

Abstract :

- - - - - - - - - - -

1 – Results should be included at the end of the abstract in terms of improvement ratio between the proposed and existing works .

Introduction Section :

- - - - - - - - - - - - - - - - - - - - - -

2 – The Introduction section is lengthy and need to be separated into Introduction and Related Work sections .

3 – organization of the manuscript should be included at the end of the Introduction section .

RGB pixel differential correction feature Section :

- - - - - - - - - - - - - - - - - - - - - - - - - - - - - - - - - - - - - - -

4 – Check for grammatical errors and typos.

Differential correction based shadow removal method Section :

- - - - - - - - - - - - - - - - - - - - - - - - - - - - - - - - - - - - - - - - - - - - - - - - - - -

5 – more visual example are required .

The structure of DC-SR method Section :

- - - - - - - - - - - - - - - - - - - - - - - - - - - - - - - - - - -

6 – Pseudo code of the method need to be included .

7 – the implementation need be included as a supplementary file for review purposes .

Experiments Section :

- - - - - - - - - - - - - - - - -

8 - Experimental computation time need to be included .

9 – Discussion is unclear .

10 – comparison with recent algorithm should be included .

11 – Evaluation metrics should be used to evaluate the performance of the proposed method .

Conclusion Section :

- - - - - - - - - - - - - - - - - - - - - -

12 – The limitation of this work should be clearly included in the conclusion section .

- - - - - - - - - - - - - - - - - - - - - - - - - - - - - - - - - - - - - - - - - - - - - - - - - - - - - - - - - - - - - - - - - - - - - - - - - - - - - - - - - - - - - - - - - - - - - - - - - - - - - - - - - - - - - - - - - - - - - - - - - - - - - - - - - - - - - - - - - - - - - - - - - - - - - - - - - - - - - - - - - - - - - - - - - - - - - - - - - - - - - - - - - - - - - - - - - - - - - - - - - - - - - - - - - - - - - - - - - - - - - - - - - - - - - - - - - - - - - - - - - - - - - - - - - - - - - - - - - - - - - - - - - - - - - - - - - - - - - -

6. PLOS authors have the option to publish the peer review history of their article (what does this mean?). If published, this will include your full peer review and any attached files.

Reviewer #1: No

---

## [Decision Letter · Decision Letter 1]

4 Oct 2022

A differential correction based shadow removal method for real-time monitoring

PONE-D-22-17507R1

Dear Dr. Liu,

We’re pleased to inform you that your manuscript has been judged scientifically suitable for publication and will be formally accepted for publication once it meets all outstanding technical requirements.

Kind regards,

Zhaoqing Pan, Ph.D.

Academic Editor

PLOS ONE

Reviewers' comments:

Reviewer's Responses to Questions

**Comments to the Author**

1. If the authors have adequately addressed your comments raised in a previous round of review and you feel that this manuscript is now acceptable for publication, you may indicate that here to bypass the “Comments to the Author” section, enter your conflict of interest statement in the “Confidential to Editor” section, and submit your "Accept" recommendation.

Reviewer #1: All comments have been addressed

2. Is the manuscript technically sound, and do the data support the conclusions?

Reviewer #1: Yes

3. Has the statistical analysis been performed appropriately and rigorously? 

Reviewer #1: Yes

4. Have the authors made all data underlying the findings in their manuscript fully available?

Reviewer #1: Yes

5. Is the manuscript presented in an intelligible fashion and written in standard English?

Reviewer #1: Yes

6. Review Comments to the Author

Reviewer #1: ID : PONE-D-22-17507 R1

Title : A differential correction based shadow removal method for real-time monitoring

Summary:

In this work, a shadow removal method-based on a differential correction is proposed. The proposed method computes the differential correlation between the pixel values of the channel in RGB color space.

In the revised manuscript, the authors have address all the raised comments .

Abstract :

- - - - - - - - - - -

1 – The abstract is fine. No comments .

Introduction Section :

- - - - - - - - - - - - - - - - - - - - - -

2 – This section is fine. No comments .

Related Work Section :

- - - - - - - - - - - - - - - - - - - - - -

3 – This section is fine. No comments .

RGB pixel differential correction feature Section :

- - - - - - - - - - - - - - - - - - - - - - - - - - - - - - - - - - - - - - -

4 – This section is fine. No comments .

Differential correction based shadow removal method Section :

- - - - - - - - - - - - - - - - - - - - - - - - - - - - - - - - - - - - - - - - - - - - - - - - - - -

5 – This section is fine. No comments .

The structure of DC-SR method Section :

- - - - - - - - - - - - - - - - - - - - - - - - - - - - - - - - - - -

6 – This section is fine. No comments .

Experiments Section :

- - - - - - - - - - - - - - - - -

7 - This section is fine. No comments .

Conclusion Section :

- - - - - - - - - - - - - - - - - - - - - -

8 – This section is fine. No comments .

- - - - - - - - - - - - - - - - - - - - - - - - - - - - - - - - - - - - - - - - - - - - - - - - - - - - - - - - - - - - - - - - - - - - - - - - - - - - - - - - - - - - - - - - - - - - - - - - - - - - - - - - - - - - - - - - - - - - - - - - - - - - - - - - - - - - - - - - - - - - - - - - - - - - - - - - - - - - - - - - - - - - - - - - - - - - - - - - - - - - - - - - - - - - - - - - - - - - - - - - - - - - - - - - - - - - - - - - - - - - - - - - - - - - - - - - - - - - - - - - - - - - - - - - - - - - - - - - - - - - - - - - - - - - - - - - - - - - - - - - - - - - - - - - - - - - - - - - - - - - - - - - - - - - - - - - - - - - - - - - - - - - - - - - - - - - - - - - - - - - - - - - - - - - - - - - - - - - - - - - - - - - - - - - - - - - - - - - - - - - - - - - - - - - - - - - - - - - - - - - - - - - - - - - - - - - - - - - - - - - - - - - - - - - - - - - - - - - - - - - - - - - - - - - - - - - - - - - - - - - - - - - - - - - - - - - - - - - - - - - - - - - - - - - - - - - - - - - - - - - - - - - - - - - - - - - - - - - - - - - - - - - - - - - - - - - - - - - - - - - - - - - - - - - - - - - - - - -

7. PLOS authors have the option to publish the peer review history of their article (what does this mean?). If published, this will include your full peer review and any attached files.

Reviewer #1: No

---

## [Editor Report · Acceptance letter]

28 Oct 2022

PONE-D-22-17507R1 

A differential correction based shadow removal method for real-time monitoring 

Dear Dr. Liu:

I'm pleased to inform you that your manuscript has been deemed suitable for publication in PLOS ONE. Congratulations! Your manuscript is now with our production department. 

Kind regards, 

on behalf of

Dr. Zhaoqing Pan 

Academic Editor

PLOS ONE